# Lipophilic Polyamines as Promising Components of Liposomal Gene Delivery Systems

**DOI:** 10.3390/pharmaceutics13060920

**Published:** 2021-06-21

**Authors:** Pavel A. Puchkov, Michael A. Maslov

**Affiliations:** Lomonosov Institute of Fine Chemical Technologies, MIREA—Russian Technological University, Vernadsky Ave. 86, 119571 Moscow, Russia; puchkov_pa@mail.ru

**Keywords:** polyamines, spermine, cationic amphiphiles, cationic liposomes, disulfide groups, gemini amphiphiles, gene delivery

## Abstract

Gene therapy requires an effective and safe delivery vehicle for nucleic acids. In the case of non-viral vehicles, including cationic liposomes, the structure of compounds composing them determines the efficiency a lot. Currently, cationic amphiphiles are the most frequently used compounds in liposomal formulations. In their structure, which is a combination of hydrophobic and cationic domains and includes spacer groups, each component contributes to the resulting delivery efficiency. This review focuses on polycationic and disulfide amphiphiles as prospective cationic amphiphiles for gene therapy and includes a discussion of the mutual influence of structural components.

## 1. Introduction

Gene therapy is a modern and promising method for treating severe hereditary and acquired diseases, including COVID-19 immunization, through the delivery of therapeutic nucleic acids (NAs) that can replace a damaged gene (pDNA), provide a new one, or block the expression of an unwanted protein (antisense oligonucleotides, siRNA) [1,2]. The direct administration of therapeutic NAs is an inefficient process due to multiple external and internal limiting factors [3]. External factors lead to the instability of NAs in biological fluids (degradation by nucleases or interaction with albumin or low-density lipoproteins, causing the aggregation and rapid clearance of NAs) and a low degree of interaction with target cells. Internal factors are determined by the presence of membrane barriers (plasma, endosomal, and nuclear membranes) that present a challenge to NAs as they attempt to reach the cytosol and nucleus [4].

Overcoming these factors requires the development of special delivery vehicles. At present, viruses [5,6], which are highly effective but present some serious disadvantages, primarily associated with the induction of inflammatory and immune responses in the body, fill this role. However, alternative non-viral delivery vehicles, such as cationic liposomes (CLs) based on cationic amphiphiles (CAs) [6,7,8,9], are being developed. The most recent success is development of an mRNA vaccine [10] against COVID-19, where lipid nanoparticles deliver nucleoside-modified mRNA encoding a mutated form of the spike protein of SARS-CoV-2 [11]. Generally, CA structure is a combination of hydrophobic and cationic domains linked together by various spacer groups [12]. The positive charge of CAs enables the “packing” of NAs due to electrostatic interactions with the formation of lipoplexes (complexes of NAs with liposomes) and facilitates their interaction with a negatively charged plasma membrane.

In addition to CAs, liposomes can include helper lipids (for example, 1,2-dioleoyl-*sn*-glycero-3-phosphoethanolamine, DOPE) [13,14,15], which promote the formation of a certain lipid phase and favor cell transfection. Liposomes may also contain additional lipophilic molecules that permit them to target certain cells [16] or increase their circulation time in the bloodstream (for example, lipophilic derivatives of polyethylene glycol, PEG) [17]. If it is not stated below, CAs were used without additional components.

The CA structure significantly affects the efficiency of NA delivery to eukaryotic cells. In recent years, many monocationic amphiphiles have been obtained [18,19,20,21,22,23,24,25,26,27,28] that form liposomes for NA delivery. In this review, we will consider polycationic amphiphiles, which, compared to monocationic analogs, enable the more efficient transport of NAs into cells due to the formation of a system of distributed charges in the polyamine matrix and their ability to facilitate NA release from endosomes. This ability is strongly affected by the high H^+^ buffer capacity of polyamines containing titratable amines results in endosomal Cl^−^ accumulation during acidification with presumed osmotic endosome disruption and enhanced lipoplex escape [29].

## 2. Cationic Amphiphiles Based on Polyamines or Amino Acids

### 2.1. Cationic Amphiphiles Based on Linear Polyamines

Enhancement of NA transport by polycationic amphiphiles may be related not only to distributed charges. On the cell surface, polyamine recognition sites—for example, PAT [30], a polyamine transporter—selectively transport both polyamines and their derivatives. Moreover, cancer cells have more such sites on their surfaces, which means that amphiphiles based on polyamines can transfect cancer cells more efficiently. Particularly important factors in NA delivery are the number and distribution of positive charges in the polyamine molecule. Transfection activity (TA) has been shown to increase as the number of amino groups in the polyamine structure increases. Compound **1e** (Figure 1) exhibited the highest transfection efficiency among the synthesized lipopolyamines **1a**–**g** [31], which suggests that **1e** can use PAT and compete with other polyamines, for example, spermine, for binding to certain recognition sites on the cell surface (in particular, with the same PAT).

Another factor affecting the efficiency of transfection is the hydrophobicity of the amphiphilic molecule. A study of compounds **2** and **3a**–**d**, which contain sterols (cortisol and its derivatives) as hydrophobic domains (Figure 2), revealed that TA increases with an increase in the hydrophobicity of the molecule [32]. While liposomes with compound **3d** were shown to be incapable of delivering NAs, possibly due to the lower hydrophobicity of the amphiphile **3d** and ineffective formation of lipoplexes, compounds **3b** and **3c** had the highest transfection efficiencies. Notably, the contribution of hydrophobicity to the efficiency of NA delivery also depends on other parameters, primarily the CA/DOPE (the last was a helper lipid) ratio and N/P ratio (the ratio of the number of CA amino groups to the number of NA phosphate groups).

Subsequent studies have shown [33] that compounds **4b** and **4c**, which contain double bonds in the polycyclic hydrophobic domain (Figure 3), were the most effective. Three-component liposomes formed from compound **4d**, its dimeric analog **4e**, and DOPE delivered plasmid DNA (pDNA) more efficiently than two-component liposomes **4e**/DOPE.

Hydrophobic substituents in the CA structure are mainly attached to primary amino groups of the polyamine. A different approach to the synthesis of CAs was proposed by Blagbrough et al. [34,35,36,37], who obtained *N*^4^,*N*^9^-disubstituted spermine derivatives **5a**–**j** with acyl or alkyl residues of various lengths and degrees of unsaturation (Figure 4). All lipoplexes formed from acyl-substituted polyamines **5a**–**h** exhibited high TA, excluding amphiphiles **5b** and **5f**. However, only compounds **5f** and **5g** with stearoyl and oleoyl residues had low toxicity toward FEK4 and HtTA cells. Notably, an increase in the degree of unsaturation of hydrocarbon chains increased both the efficiency of transfection and the cytotoxicity of the compounds. Alkyl derivatives of spermine **5i** and **5j** had comparable or slightly higher transfection efficiencies but were much more toxic than their acyl analogs [35].

Asymmetric analogs **6a**–**f** (Figure 4) were subsequently developed [38]. *N*^4^-myristoleoyl-*N*^9^-myristoylspermine (**6c**) and *N*^4^-oleoyl-*N*^9^-stearoylspermine (**6d**) showed the highest efficiency of siRNA delivery into FEK4 and HtTA cells, comparable to the efficiency of the commercial transfectant TransIT-TKO (Mirus Bio, Madison, WI, USA). Amphiphile **6f** with a lithocholoyl residue effectively delivered NAs but caused cell death. The least effective was *N*^4^-cholesteryl-*N*^9^-oleoylspermine (**6e**).

When *N*^1^,*N*^12^-substituted spermine derivatives **7a**–**d** (Figure 5), structural isomers of amphiphiles **5c**, **5d**, **5f**, **5g**, were synthesized and studied [39], the efficiency of pDNA delivery into FEK4 and HtTA cells by complexes with amphiphiles **7a**–**d** was lower, while the toxicity was higher than with amphiphile **5g**. siRNA delivery efficiency mediated by compounds **7a** and **7c** was comparable to the efficiency of delivery using amphiphile **5g**.

Multiple monosubstituted polyamine derivatives **8a**–**g** (Figure 5) were obtained by modifying spermine with fatty acid residues of various lengths and degrees of unsaturation [40]. Although an increase in the length of the fatty acid residue increased toxicity, it positively affected the penetration of lipoplexes through the cell membrane in vitro. Experiments in vivo showed that the efficiency of NA delivery with *N*-butanoylspermine (**8f**) was higher than with *N*-decanoylspermine (**8e**).

Mono- and disubstituted polycationic amphiphiles were developed based on spermine as a hydrophilic domain and cholesterol or 1,2-di-*O*-tetradecylglycerol as hydrophobic domains (Figure 6) [41,42,43]. The amphiphiles had different spacer lengths and linker types. CLs were prepared using these amphiphiles and DOPE (1:1 mol.). Among monosubstituted amphiphiles **9a**–**c**, compound **9b** showed the highest TA. While transfecting the same percentage of cells as their monomeric analogs **9a**–**c**, however, dimeric polycationic amphiphiles **10a**–**c** provided better expression of the green fluorescent protein. The highest transfection efficiency was exhibited by liposomes based on amphiphile **10c**, which were superior to the efficiency of the commercial transfectant Lipofectamine 2000 (Thermo Fisher Scientific, Waltham, MA, USA) for any type of NAs transferred [42,43]. Targeted liposomes based on CA **10c** were also successfully employed in vivo [44,45,46,47,48,49].

Analogs **10e**–**g** with ethoxypropylene, octamethylene, and ethoxyethoxyethylene spacers (Figure 6) permitted a greater TA increase than using **10c** in vitro [50]. In contrast, the replacement of spermine with triethylenetetramine (TETA, **11a**,**b**) led to a significant decrease in TA [51].

Extensive screening of CAs [52] revealed that in the structure of compounds **12a**–**j** through **29a**–**j**, both the polyamine matrix and the hydrophobic components changed (Figure 7). Among CLs formed from these amphiphiles and DOPE (1:2 weight ratio), the effective transfection of HEK293 cells was achieved only by liposomes with amphiphiles **12a**–**j** through **20a**–**j** containing an acyl substituent at the terminal amino group. Moreover, only eight compounds (**12c**, **12e**, **13d**, **14c**, **16d**, **16g**, **17h**, and **17j**) were superior in TA to the commercial transfectant Effectene (Qiagen, Hilton, Germany). Subsequent transfection studies on HEK293, COLO 205, D17, HeLa, and PC3 cells showed that these compounds mediated more effective NA transport than did the commercial transfectants Effectene, DOTAP, and DC-Chol, while their toxicity was lower than that of commercial transfectants.

pH-Sensitive polycationic amphiphiles **30**–**33** (Figure 8) were obtained by subsequently coupling amino acids (l-histidine and l-cysteine) and fatty acids (lauric, oleic, and stearic) to polyamines [53,54]. The size of complexes of amphiphiles with siRNA was 160–210 nm, and the maximum TA on U87 cells was achieved using amphiphiles **30b**–**33b** with oleic residues. Among them, TA decreased in the series **30b ** >  **32b ** >  **31b ** >  **33b**. A correlation was also established between the TA and the ability of compounds **30b**–**33b** to disrupt the integrity of erythrocyte membranes. Leader compound **30b** based on ethylenediamine exhibited the highest hemolytic activity at pH 5.4, which corresponds to the onset of endosomal acidification. Therefore, when using this amphiphile, one can expect effective NA release inside cells due to the disruption of endosomal membranes.

The second generation of pH-sensitive amphiphiles **34a**–**h** based on spermine was subsequently obtained (Figure 9). In biological tests conducted on HeLa and U87 cells, the presence of an l-histidine in the amphiphile structure did not improve TA. In addition, no relationship was found between the efficiency of CAs and the distance between hydrophobic domains. Compound **34e** exhibited the highest activity in the delivery of pDNA [55], while amphiphile **34f** exhibited the highest activity in the delivery of siRNA [56,57]. The authors also noted that they did not utilize helper lipids in complex formation since the synthesized compounds were able to initiate a pH-dependent phase transition, which led to the destabilization of the complexes and the release of NAs.

Multiple phosphamide derivatives containing long-chain alkyl substituents (dodecyl, tetradecyl, and hexadecyl) were obtained as hydrophobic fragments [58]. The transfection of COS-1 cells with lipoplexes formed by pDNA and micelles or liposomes based on amphiphiles **35a**–**d** (Figure 10) showed that complexes based on micelles were only half as effective as complexes based on CA liposomes/lipid helper/Chol (1:1:1 mol.). DOPE and dipalmitoyl phosphatidylcholine (DPPC) have been used as helper lipids, but DPPC-containing liposomes have proven to be an ineffective delivery vehicle. TA on LLC and B16BL6 cells increased with an increase in the length of the alkyl chains and the number of amino groups in the polyamine. For LLC cells, the best compound was **35f** based on spermine, and for B16BL6 cells, the best compound was amphiphile **35c** based on spermidine.

An analog of compounds **35c** and **35f** was obtained based on a synthetic polyamine–tetraethylenepentamine (**35g**, Figure 10) [59]. Liposomes **35g**/DOPE/DPPC/Chol (0.25:1:0.75:1 mol) efficiently delivered antisense oligonucleotides to eukaryotic cells. Here, the introduction of lipophilic derivatives of polyethylene glycol (PEG) and a cyclic analog of the peptide RGD ensured active targeting of liposomes to target cells and increased the efficiency of NA delivery [60,61].

Cationic nucleoside amphiphiles may also be used for gene delivery. Thus, low-toxic uridine derivatives of various polyamines (**36a**–**c**, Figure 11) were synthesized and used for siRNA delivery. Their TA on HeLa cells is almost equal to that of Lipofectamine 2000 but was not affected by polyamine residue [62]. Notably, replacement of polyamine residue with L-arginine gave the same results, while l-lysine decreased TA [63].

Amphiphiles **37a**–**d** (Figure 12), in which the polyamine was bound to the hydrophobic domain via carbamoyl or amide linkers, formed liposomes with DOPE or compound **35h** (Figure 11) and were used to deliver pDNA [64]. Protonation of the imidazolium residue of amphiphile **35h** during endosomal acidification can induce rupture of the endosomal membrane and favors NA release [65,66]. Transfection of OVCAR-3, IGROV-1, and HeLa cells with complexes formed at different N/P ratios (4:1–12:1) showed that **37c**/DOPE liposomes provided efficient pDNA delivery exceeding that of the commercial transfectant Lipofectamine 2000. Relative TA decreased in the series **37c** > **37b** > **37a** >> **37d**. It should also be noted that the use of amphiphile **36** as a helper lipid did not increase TA but did increase the cytotoxicity of lipoplexes.

In vivo delivery of pDNA by sterically stabilized liposomes **37c**/DOPE/PEG4600-Chol (43:43:14 mol.) in a 4:1 (wt) ratio with pDNA led to a 33-fold increase in protein expression relative to unprotected DNA [67].

New CAs **38a**–**c** (Figure 13), in which the cationic domain was linked to the cholesterol residue via an ether bond [68], formed liposomes with DOPE and were used for transfection of AGS and Huh-7 cells. The **38a**/DOPE liposomes more efficiently delivered pDNA into AGS cells, while the **38b**/DOPE liposomes provided effective transfection of Huh-7 cells. In both cases, their TA exceeded that of commercial transfectants [69]. Liposomes with dimeric gemini-amphiphile **38c** also outperformed commercial agents in the transfection of COS-7 and Huh-7 cells [70].

CAs **39a,b** with different dicationic domains (Figure 14) formed liposomes, which facilitated the transport of siRNA into MB49 and K562 cells, while amphiphile **39a** was superior in TA to amphiphile **39b** [71].

Comparing the TA of CAs that contained various polyamines in their structure (Figure 15) revealed that CLs composed of both phosphatidylcholine (Phospholipon 90G) and compounds **40d**–**g** containing spermine (5:1 mol.) could deliver pDNA to HeLa cells, while other CLs showed no transfection [72]. Moreover, CAs with a shorter chain length of acyl substituents exhibited lower TA.

Analogs of compounds **40d**–**f** based on spermine (compounds **41a**–**c**, **42a**–**c**, **43a**–**c**) were obtained to study the influence of the structure core on TA (Figure 16) [73]. The efficiency of pDNA delivery mediated by liposomes based on DOPE and amphiphiles **42b**,**c**, and **43a** (1:1, weight ratio) was higher or comparable to that of Lipofectamine 2000. Moreover, unlike the other formulations, liposomes based on **43a** with a core of 2-amino-1,3-propanediol retained their efficiency in the presence of serum. Investigation of the effect of hydrophobic domains on transfection revealed that myristoyl residues provided more effective TA.

A library of CAs (more than 1200 compounds) was developed using combinatorial chemistry methodology, in which both the hydrophobic (the length of the alkyl chain, the type of linker, and the presence of additional functional groups) and the cationic (the number of amino groups, the presence of cycles, and other functional groups) domains varied [74]. The results of in vitro experiments on HeLa cells revealed the following relationships: (1) TA increased in the presence of either two long-chain or several shorter alkyl substituents linked by an amide bond to the cationic domain (the optimal length was 8–12 carbon atoms); (2) high TA was achieved by compounds with two or more amino groups in the cationic domain, with TETA offering the best option; (3) the presence of a secondary amino group in the cationic domain positively affected TA (Figure 17).

Based on these findings, multiple CAs were selected for extended biological studies, which showed that the efficiency of NA delivery to primary macrophages exceeded that of commercial transfectants. In contrast, the transfection of HeLa or HepG2 cells by the selected amphiphiles was poor.

According to the results of in vitro tests, the 17 most effective compounds for in vivo siRNA delivery (siFVII and siApoB, which suppress the expression of blood coagulation factor VII and apolipoprotein B, respectively), were selected. For this, liposomes containing CA/PEG-lipid (mPEG2000-palmitoylceramide or mPEG2000-dimyristoylglycerol)/Chol (42:10:48 mol.) were formed. These CAs commonly contained various diamines or TETA. The most effective formulation (achieving more than 90% suppression of target gene expression) was based on compound **44** with TETA (Figure 17). The delivery of siRNAs in the lungs and macrophages of mice and macaque liver cells using these liposomes also significantly suppressed the target genes [74].

Subsequently, the library of compounds was expanded by synthesizing amphiphiles **45а**–**d** and **46а**–**d** with various hydrophobic domains (TETA and propylenediamine were chosen as cationic domains). Of these, the most effective were amphiphiles with a hydroxyl group in the hydrophobic domain, while the domain itself was bound to the polyamine with an ether or amide linker [75]. The hydrophobic domains based on oligoethylene glycol or octadecyl substituents led to an absence of TA. It should be noted that CAs **45c**, **46a**, and **46d** were more effective than amphiphile **44** in vitro; however, they were inferior to amphiphile **44** in siRNA delivery in vivo.

In vitro screening of amphiphilic derivatives of spermine, spermidine, putrescine, and cadaverine showed that spermine derivative **47e** facilitated the more efficient delivery of siRNA into human hepatocellular carcinoma Huh-7 cells [76]. Furthermore, liposomes **47e**/Chol/DSPC/mPEG2000-palmitoylceramide/galactosylceramide delivered siRNA in vivo, which led to a significant decrease in hepatitis C virus replication in the hepatocyte cells of mice [76].

### 2.2. Cationic Amphiphiles Based on Cyclic Polyamines

Lipophilic derivatives of macrocyclic polyamines can be used as liposomal components. Such compounds are less prone to self-packing, which improves binding to NAs. Two new cyclen-based amphiphiles **48a,b** (Figure 18), which differed in the type of hydrophobic domain (cholesterol or diosgenin), were synthesized [77]. The results of in vitro biological studies showed that liposomes formed by amphiphiles **48a,b** and DOPE had a low cytotoxicity but, because they bound easily to serum proteins, were inferior to Lipofectamine 2000 in delivering NAs to cells. The introduction of the quaternary ammonium group into the structure of the amphiphile (amphiphiles **49a,b**) did not increase the TA; the liposomes remained inferior to Lipofectamine 2000 [78]. However, amphiphile **49b**, which contained diosgenin as a hydrophobic domain, was more effective but also more toxic than amphiphile **49a**, which contained cholesterol.

In order to increase the TA of liposomes, compounds **50a**–**c** were synthesized (Figure 18), in which the cyclene and the hydrophobic domain were arranged by a peptide–nucleoside spacer [79]. Amphiphile **50b** with a diosgenin residue (in liposomal composition with DOPE) demonstrated the highest efficiency, exceeding that of Lipofectamine 2000.

In the structure of CAs **51a**–**e**, hydrophobic domains were linked to the cyclen using the l-histidine residue (Figure 19), as well as additional linkers (amide, ether, carbamoyl, and ester) [80]. All lipoplexes formed by amphiphiles **51a**–**e**, DOPE and NAs were less toxic than Lipofectamine 2000, and amphiphile **51a** with hexadecyl substituents demonstrated the lowest toxicity. The results of transfection studies revealed that only amphiphiles **51c**–**e**, containing a tocopherol residue as a hydrophobic domain, were more effective than Lipofectamine 2000. Compound **51e** with an ester linker exhibited the highest TA.

Cyclen derivatives containing oleoyl residues as a hydrophobic domain and amino acids (l-phenylalanine, l-tyrosine, l-serine, or glycine) as a backbone (Figure 20) were developed [81]. Transfection of HEK293 cells showed that liposomes based on DOPE and compounds **52a**–**d** with one fatty acid residue (1:1 mol.) exhibited no TA. Among amphiphiles with two long-chain substituents, the compounds with phenyl (**53**) or phenylalanyl (**55a**) spacers demonstrated the highest activity (which was nevertheless lower than that of Lipofectamine 2000). Moreover, analogs with saturated hydrocarbon residues of various lengths were obtained based on amphiphile **53** (structures not shown). Among them, an amphiphile with tetradecyl substituents was the most effective, comparable to Lipofectamine 2000.

### 2.3. Cationic Amphiphiles Based on Amino Acids

Amino acids are often employed as the hydrophilic domains of CAs, as they are natural compounds and convenient building blocks for creating a positive charge in molecules. To identify new compounds for CL formulation, a number of amphiphiles **56**–**58** (Figure 21) based on l-lysine, l-histidine, and l-arginine was synthesized and further used with DOPE (1:1 mol.) for liposomal preparations. Their efficiency of NA delivery in the presence of serum was studied [82].

TA is known to decrease in the presence of serum due to the interaction of negatively charged proteins with CAs, which inhibits NA delivery. The average size of lipoplexes formed by amphiphiles **56**–**58** increased in the presence of serum from 180 nm to 828 nm for amphiphile **56**, up to 1710 nm for amphiphile **57**, and up to 2345 nm for amphiphile **58**. Amphiphile **56** was able to transfect eukaryotic cells at serum concentrations up to 40%. The TA of lipoplexes containing amphiphiles **56**–**58** was 2.8–3.5-fold higher than that of the commercial transfectant DOTAP (Roche Life Technologies, Switzerland), while amphiphile **57**, with an L-histidine residue, exhibited the lowest activity.

In the structure of CAs **59a**,**b** and **60a**,**b** (Figure 22), cholesterol was bound to l-lysine or l-histidine by various linker groups [83]. The best transfection of COS-7 cells was provided by amphiphiles **59b** and **60b** with l-lysine, which, however, demonstrated greater cytotoxicity than amphiphiles **59a** and **60a** containing l-histidine. The authors proposed that this greater toxicity was induced by the damaging electrostatic interaction of l-lysine amino groups with the negatively charged plasma membrane. Replacing the ester linker with an amide one (amphiphiles **61a**,**b**) reduced the toxicity of the compounds. The l-lysine derivative **61b** was still more effective than amphiphile **61a** based on l-histidine [84].

Amphiphiles **62a**–**d** are based on l-arginine and certain sterols linked by amide and ester bonds (Figure 23) [85]. Cholesterol-based compound **62а** exhibited the highest TA, forming lipoplexes with a particle size of approximately 100 nm. The size of lipoplexes formed by amphiphiles **62c**,**d** was 200–300 nm, and they poorly penetrated cells. It should be noted that none of the amphiphiles **62a**–**d** was able to outperform Lipofectamine 2000 in the presence of serum.

CAs **63a**–**c** and **63e**–**j** (Figure 24) were synthesized based on l-arginine, l-lysine, and l-histidine [86]. The efficiency of cell transfection decreased with an increase in the length of the alkyl chains, while the maximum efficiency was provided by amphiphiles **63a** and **63e** with tetradecyl substituents. The type of the cationic group insignificantly affected TA in the absence of serum. However, amphiphiles **63a**–**c** based on L-arginine demonstrated the most efficient NA delivery in the presence of serum.

An increase in the number of positively charged groups in amphiphiles **63k**,**l** provided an increase in the efficiency of pDNA delivery to HepG2 cells. Moreover, the resulting TA was superior to that of both Lipofectamine 2000 and polyethyleneimine [87]. For HEK293T cells, amphiphile **63k** based on l-arginine was comparable to commercial transfectants, while the l-lysine derivative **63l** demonstrated inferior efficiency. The results of pDNA delivery in vivo showed that CA **63l** was 2.5-fold more effective than polyethyleneimine, while **63k** was almost 10-fold more effective.

Low-toxicity lysine-containing CAs **63d**, **64a**–**h** (Figure 24) were synthesized to study the effect of spacer length and the type of counterion on TA [88]. CLs formed by DOPE and trifluoroacetates **64g** and **64h** delivered NAs more efficiently than other CAs, while CLs containing uncharged amphiphiles **64a**,**b** demonstrated the lowest activity. An increase in the spacer length in CAs from 0 to 7 methylene units increased TA in both the presence and absence of serum, and all amphiphiles studied were more effective than Lipofectamine 2000 under both conditions.

A similar effect of spacer length on TA was observed in the amphiphile series **63f**, **64k**–**m**, and **65** based on l-lysine (Figure 24) [89]. Lipoplexes with amphiphile **63f**, the structure of which lacks a spacer, were unable to deliver NA, but as the spacer length increased from 3 to 7 methylene units, TA increased. Notably, amphiphiles **64k**–**m** with a hydrophobic oligomethylene spacer functioned as more active transfectants than amphiphiles **65** containing a hydrophilic oxyethylene spacer.

Amphiphiles **64d**,**g** and **64i**,**j** based on l-lysine and l-arginine (Figure 24) were studied to evaluate TA in PC-12, HeLa, and neuronal cells [90]. Lipoplexes containing amphiphiles **64i**,**j** were more effective than both lipoplexes containing amphiphiles **64d**,**g** and the commercial transfectant Lipofectamine 2000.

Cholesterol-containing CAs **66a**,**b** (Figure 25) were recently developed based on l-lysine and diamines [91]. When transfecting HEK293, HeLa, PC-3, and HC-04 cells, the efficacy of liposomes 66b/DOPE (1:1, mol.) was comparable to that of Lipofectamine 2000, even in the presence of serum. Moreover, the toxicity of 66b/DOPE was significantly lower than that of Lipofectamine 2000.

Polycationic amphiphiles **67а**,**b** (Figure 26) consisting of the dipeptides glycylhistidine and glycyllysine as well as a tocopherol residue were synthesized [92]. While the TA of **67b**/DOPE liposomes containing a lysine residue was higher on HEK293 and HeLa cells than that of **67a**/DOPE liposomes containing a histidine residue, the **67b**/DOPE liposomes did not achieve the results of Lipofectamine 2000. However, the complexes were more resistant to serum activity than was the commercial transfectant.

Polycationic amphiphiles **68a**–**d** and **69d** (Figure 27) were synthesized based on l -ornithine [93,94]. Liposomes CA/DOPC and complexes with pDNA were formed using different component ratios. The resulting efficiency of cell transfection, which was higher than for the commercial transfectant DOTAP (Avanti Polar Lipids, Alabaster, AL, USA), depended on the ratio of lipids used in the complex. Amphiphile 68d with three aminoethyl substituents exhibited the highest activity and was also used to deliver siRNA, which led to the specific suppression of the target gene [95].

Amino acids are used to develop not only classical head-to-tail amphiphiles but also symmetric gemini-amphiphiles, as seen in the l -histidine derivatives **70a**–**d** (Figure 28), which contain two alkyl substituents 10 to 16 carbon atoms in length [96,97]. CLs were formed with derivatives obtained and DOPE. With the exception of **70a** with decyl residues, all CLs bound pDNA well. However, their TA was comparable to or lower than that of Lipofectamine 2000 [96].

### 2.4. Disulfide Cationic Amphiphiles

Several special cell proteins and peptides (for example, glutathione and reductases) can reduce the disulfide bond [98]. Disruption of the disulfide bond in the CA molecule, in turn, can increase the degree of NA release from the lipoplex in the cell and thereby increase TA.

Disulfide amphiphile **71**, based on l-lysine and l-arginine (Figure 29), formed CLs capable of transfecting HeLa and B16 cells in the absence or presence of serum, and its efficiency exceeded that of its analog lacking a disulfide bond [99]. In vivo studies on the delivery of the luciferase gene to xenograft mice showed a high expression of the reporter protein in isolated tumors in the case of CA **71**, which exceeded that of the amphiphile without a disulfide bond (structure not shown) or polyethyleneimine.

During the oxidation of thiol groups to disulfide groups, lipophilic thio compounds **72a**–**d** (Figure 30) based on l-ornithine or spermine compacted pDNA molecules into small micellar complexes [100]. Their TA in 3T3 cells increased with increasing alkyl substituents. Spermine-based CA **72d** was less effective than its ornithine-based analog **72b**.

Disulfide amphiphiles **73a**,**b** (Figure 31) containing an l-histidine or l-lysine residue were obtained based on tocopherol [101]. Liposomes CA/DOPC (1:1 mol) formed complexes with pDNA at various N/P ratios (from 1 to 8) with sizes ranging from 150 to 190 nm. The TA of liposomes **73b** was lower than that of liposomes based on l-histidine derivative **73a** and the commercial transfectant Lipofectamine 2000.

The introduction of a disulfide bond into the amphiphile structure also makes it possible to significantly reduce CAs toxicity while maintaining a comparable level of TA. CA/DOPC liposomes, based on biodegradable CAs **74a**–**d** (Figure 32) with a disulfide bond, differed in the number of amino groups in the polyamine matrix had significantly lower toxicity (except for amphiphile **74a**) than their analogs **68a**–**d**, which did not contain a disulfide bond [102]. Moreover, compounds **74c** and **74d** demonstrated comparable or superior TA to amphiphile **68d**. Thus, an increase in the number of amino groups in CAs increased TA, while the introduction of a disulfide group decreased the toxicity of lipoplexes, allowing for the use of higher CA dosages.

Biological testing on COS-7 cells of disulfide amphiphiles **75a**,**b** (Figure 33), based on polyamines and amino acids [103], revealed low toxicity and rather high TA comparable to monocationic analogs **75c**,**d**. Moreover, only amphiphile **75a** retained its activity in the presence of serum.

Synthesized gemini-amphiphiles **76a**–**c** (Figure 34), which contained disulfide groups between the cationic and hydrophobic domains, differed in the spacer connecting the quaternary ammonium groups [104]. Liposomes **76a**/DOPE and **76c**/DOPE efficiently transfected HeLa and HT1080 cells. For PC3AR cells, liposomes **76b**/DOPE proved to be the most efficient and were also capable of delivering NAs to difficult-to-transfect HaCaT cells.

Disulfide polycationic amphiphiles **77a**,**b** (Figure 35) were based on spermine and cholesterol. The disulfide group was located either in the spacer structure or as a linker connecting the cationic and hydrophobic domains [105,106]. CLs were formed with these amphiphiles and DOPE. The position of the disulfide group in the CA molecule affected both the sensitivity of the obtained liposomes to the action of reducing agents and their TA. Thus, introducing a disulfide group into the spacer structure (**77a**) led to a higher liposome sensitivity and more efficient siRNA delivery than that achieved by CA **10c** (Figure 6), which was not sensitive to reduction. CA **77b** with a disulfide group as a linker was less sensitive to reduction and delivered pDNA more efficiently than did its analogs [107].

A library of disulfide CAs **78**–**89** (Figure 36) with various cationic domains was created. The first series of compounds, **79a**–**84a**, contained a hydrophobic hexadecyl chain, including sulfur atoms [108]. CLs composed of disulfide CAs, cholesterol, DOPE and 1,2-distearoyl-*sn*-glycero-3-phosphoethanolamine-*N*-[methoxy(polyethylene glycol)-2000] (16:4:1:1, weight ratio) were used to deliver ribonucleoprotein complexes to GFP-HEK cells. Compounds **80a**, **81a**, **83a**, **84a** had the highest efficacy, comparable to or exceeding that of Lipofectamine 2000. An increase in the number of hydrophobic domains to three (**78b**–**81b**) increased the delivery efficiency of ribonucleoprotein complexes to GFP-HEK cells [109] and human mesenchymal stem cells [110]. Notably, liposomes based on compound **79a** with two hydrophobic domains had practically no TA, while the use of an analog (**79b**) with three hydrophobic domains made it possible to transfect stem cells more efficiently than was possible with the commercial transfectant Lipofectamine 2000. An increase in the number of hydrophobic domains to four negatively affected TA regardless of the cationic domain. A study of the effect of the length of the hydrophobic domain demonstrated that shortening the chain length to 12 atoms significantly impaired TA, while including 14–18 atoms in the chain provided no efficiency difference [110].

Subsequently, the library of disulfide derivatives was expanded using a cholesterol residue in the structure of the hydrophobic domain, which reduced cytotoxicity relative to that of long-chain analogs **79a**–**84a** [111]. Compounds **85c**, **87c**, **88c** were highly efficient in delivering mRNA to all four investigated cell lines (B16F10, HEK, HeLa, NIH 3T3). Compound **86c** was effective on all cells except NIH 3T3. The results of all four compounds were comparable or slightly inferior to that of Lipofectamine 2000. Compounds **84c** and **89c** demonstrated extremely low aggregation stability and delivery efficiency.

### 2.5. Influence of Structural Components of Cationic Amphiphiles on the Efficiency of Nucleic Acid Delivery

Each element of the CA structure performs a specific function and influences the TA. Hydrophobic domains are involved in the protection of NAs and promote the fusion of lipoplexes with cell membranes. Aliphatic hydrocarbon substituents usually represent these domains with a length of 10 to 18 carbon atoms, tocopherol, or sterols. The type of hydrophobic domain determines both the structure of the vesicles that a CA forms in the aqueous phase and its subsequent interaction with biological membranes. Liposomal formulations promote more efficient NA delivery than that accomplished by micelles or other types of nanoparticles [58]. Mostly, CAs used for transfection of eukaryotic cells are classic head-to-tail amphiphiles. Based on polyamines and amino acids, CAs synthesized with two hydrophobic domains (gemini-amphiphiles) can deliver NAs more efficiently than can their monosubstituted analogs [39,42,94,95].

An increase in the length of aliphatic hydrocarbon chains usually increases TA [32,35,40,53,58,88,89]. Notably, however, it may also increase the toxicity of compounds [40]. In contrast, for some spermine-based CAs, TA decreased with an increase in the length of aliphatic substituents [70,84]. Analysis of published data reveals that the optimal length of aliphatic substituents is 14–18 carbon atoms, while high TA is most often noted for CAs with myristoyl or tetradecyl substituents [71,79,84]. The degree of unsaturation of substituents also affects the efficiency of NA delivery: with an increase in unsaturation, TA increases but so does toxicity [20]. Thus, it is necessary to search for an optimal CA variant that effectively delivers NAs, while its toxicity remains within acceptable limits.

When sterol derivatives are used as hydrophobic domains, one should prefer natural compounds, which do not cause significant toxicity. In this case, it is optimal to use a common and widely available sterol such as cholesterol [38,78,85], although diosgenin derivatives are also capable of efficient NA delivery [75,76,77,83]. Tocopherol is also employed as a hydrophobic domain [80,92,101].

The positively charged CA domain is responsible for the electrostatic interactions of amphiphiles and/or liposomes with NAs, the formation of stable lipoplexes, and the interaction of complexes with cell membranes. An increase in the number of amino groups in the structure of the cationic domain leads to an increase in the TA [31,56,60,81,82,85]. The most effective are CAs with domains based on polyamines, with the number (more than two [74]) and distribution of amino groups in the polyamine chain [31,53,93,102] playing important roles. Many studies reveal that the most effective are polyamines with four amino groups, primarily a natural polyamine—spermine [58,72,76]. However, CAs based on synthetic polyamines (TETA, triethylenepentamine, tributylenepentamine) can be more efficient in delivering NAs than their natural counterparts [31,59,74]. The cationic domain can also be designed based on cyclen [77,78,79,80,81] or amino acids. In this case, it is advisable to use the residues of L-arginine or l-lysine [82,83,84,86,90,92] but not l-histidine, the presence of which in the CA structure did not improve the NA delivery [84].

Linkers, connecting hydrophobic and cationic domains, determine the stability and biocompatibility of the CAs and play a key role in the efficiency of NA delivery. The most commonly used are ether, ester, carbamate, amide, and disulfide linkers. Among the most effective linkers imparting low toxicity to CAs are carbamoyl ones [42,43,52,87]. Efficient delivery of NAs is also observed with the presence of ether bonds in the CA structure [68,82]. The introduction of a disulfide linker into a CA molecule makes it sensitive to the action of intracellular reducing agents, which can increase the efficiency of NA delivery and/or reduce the toxicity of compounds [101,102,104].

Multiple studies have proven that a close arrangement of the hydrophobic and cationic domains complicates both domain’s functioning and interferes with the formation of liposomes. The introduction of a spacer into the CA structure and an increase in its length increase NA delivery efficiency [42,50,88,89,91].

Low cytotoxicity is an important feature of CLs. First generation of CAs containing quaternary ammonium head was rather toxic, but numerous recently developed polycationic amphiphiles provided non-toxic transfection [8]. As mentioned above, toxicity may be increased with the length and degree of unsaturation of hydrophobic tails. In sterol derivatives use of diosgenin may cause toxicity [77,85] probably due to the inhibition of 3-hydroxy-3-methylglutaryl coenzyme A (HMG-CoA) reductase. Linkers may also affect toxicity. Thus, the replacement of the ester linker with an amide one reduced the toxicity of the L-lysine based CAs [84]. Also, it should be noted that direct binding of both hydrophobic and cationic domains leads to a significant increase in toxicity of the CAs [35]. 

## 3. Conclusions and Perspectives

In this review, we focused on the cationic amphiphiles, which are the major but not sole component of CLs. Development of additional components, such as targeting [16] or stealth [17] lipids, is an actual trend in non-viral gene delivery.

Positive charge of CLs provides an effective binding and protection of NAs during transfection but also attracts serum proteins and other components during in vivo administration. Interaction with serum components may cause fast blood clearance of lipoplexes. Therefore, searching for possibilities to mask excessive CLs positive charge is required. One of such ways may be a use of PEG-lipids, which, however, decrease both lipoplex and NAs internalization into the cell. Alternatively, a coating of CLs or lipoplexes with additional lipid (polymer) shell may be also considered.

Sophistication of liposomal formulations turns its manufacturing procedure into novel challenges including scale-up and quality control stages. A possible solution to overcome existing difficulties might be a shift from the single large-scale to the multiple parallel small-scale production. For this purpose, microfluidic technologies become very attractive due to their ability to provide continuous and reproducible liposome preparation. Recent progress in microfluidics allows to use this technique for production of rather complex nanoparticles containing biopolymers and even whole cells [112].

In conclusion, the development of effective and safe CLs for the delivery of therapeutic NAs requires employment of the right combination of structural elements in the CA molecule to promote the formation of both the liposomes and their complexes with NAs while avoiding interference and overcoming biological barriers. Once within the target cell, the complexes must release NAs with a high efficiency to provide biological/therapeutic effect.

## Figures and Tables

**Figure 1 pharmaceutics-13-00920-f001:**
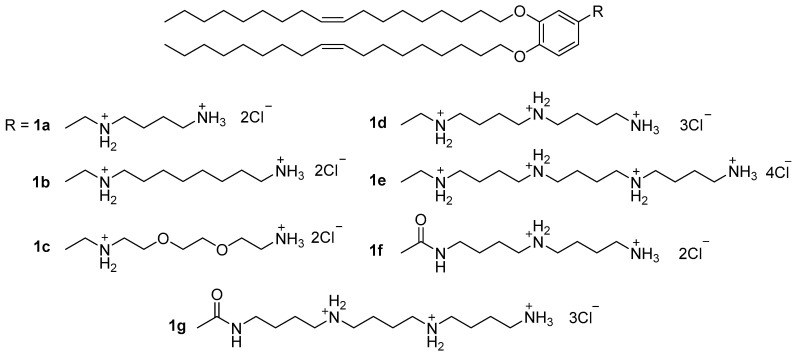
Lipopolyamines with benzyl linker.

**Figure 2 pharmaceutics-13-00920-f002:**
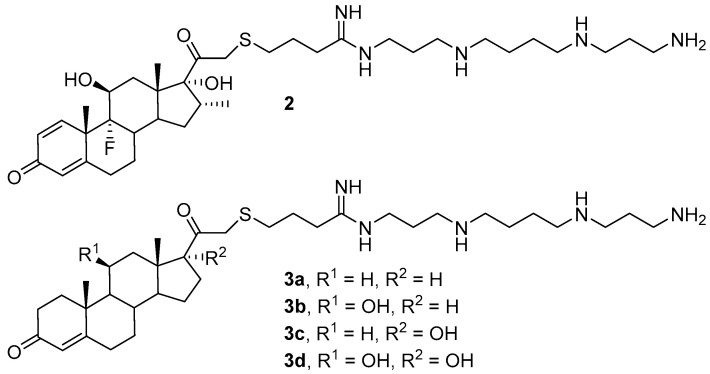
Cationic amphiphiles (CAs) based on cortisol and its derivatives.

**Figure 3 pharmaceutics-13-00920-f003:**
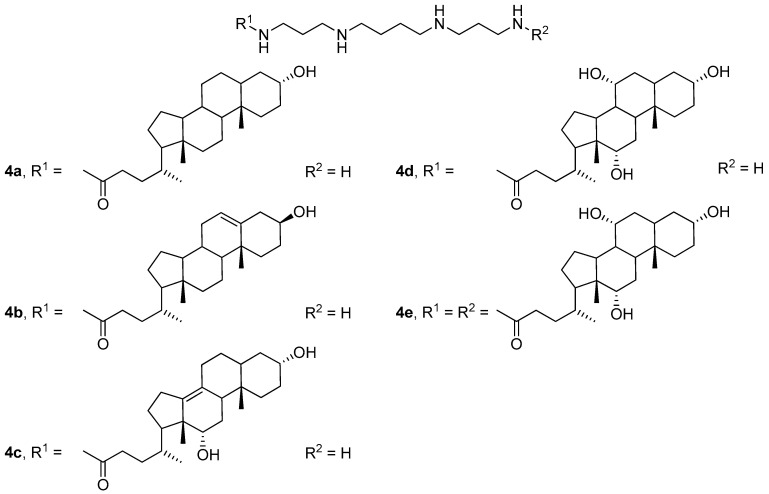
CAs with different polycyclic hydrophobic domains.

**Figure 4 pharmaceutics-13-00920-f004:**
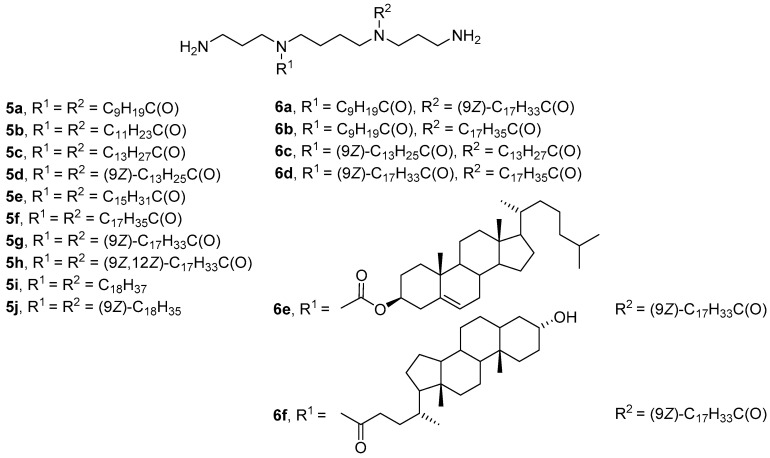
*N*^4^,*N*^9^-disubstituted spermine derivatives.

**Figure 5 pharmaceutics-13-00920-f005:**
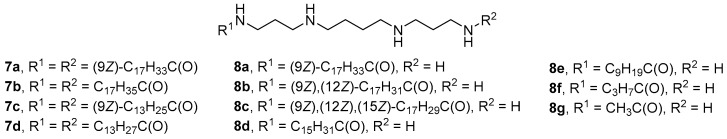
Spermine derivatives with acyl-substituted terminal amino groups.

**Figure 6 pharmaceutics-13-00920-f006:**
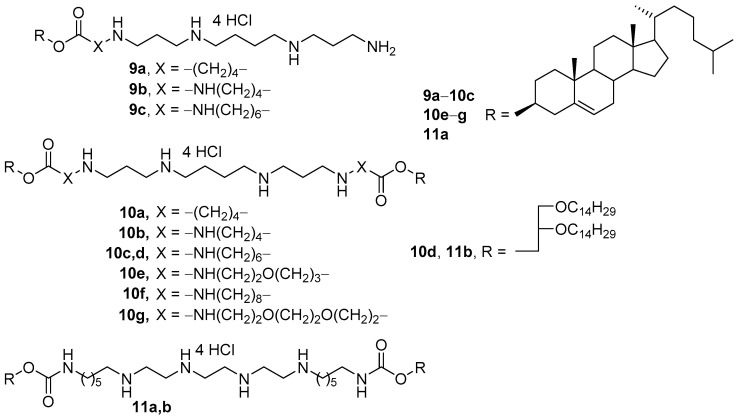
Mono- and dimeric polycationic amphiphiles based on spermine and triethylenetetramine.

**Figure 7 pharmaceutics-13-00920-f007:**
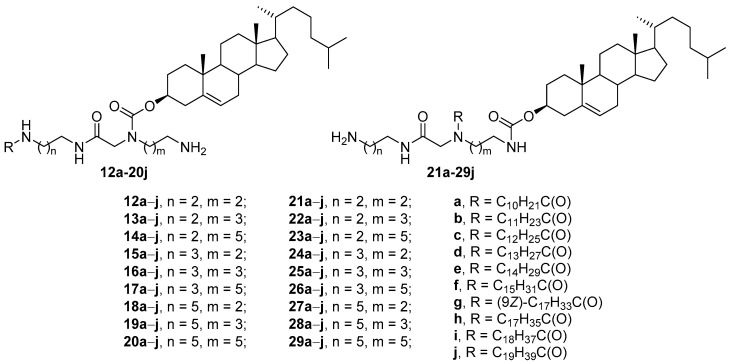
CAs with asymmetric acyl hydrophobic tails.

**Figure 8 pharmaceutics-13-00920-f008:**
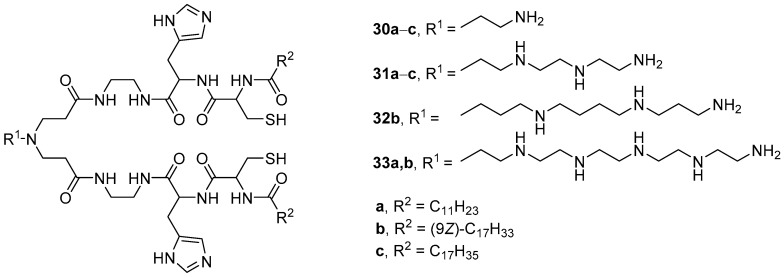
First generation of pH-sensitive polycationic amphiphiles.

**Figure 9 pharmaceutics-13-00920-f009:**
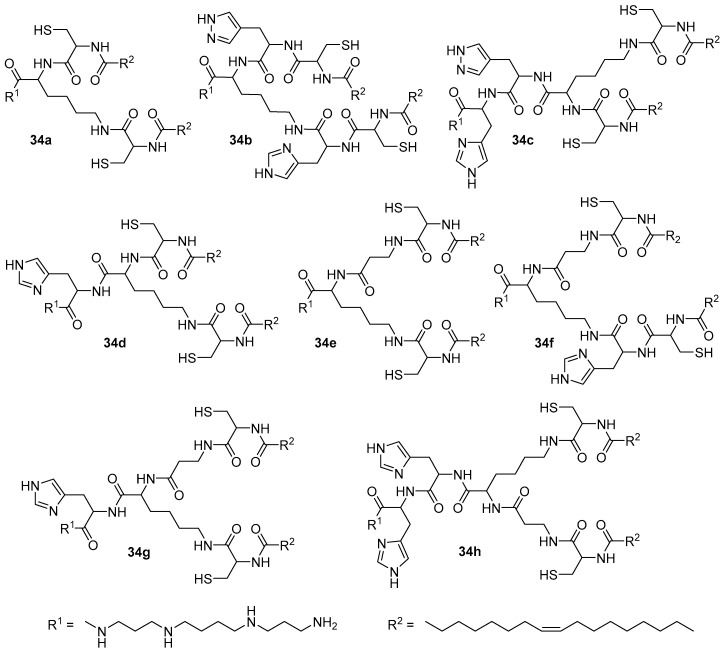
Second generation of pH-sensitive polycationic amphiphiles.

**Figure 10 pharmaceutics-13-00920-f010:**
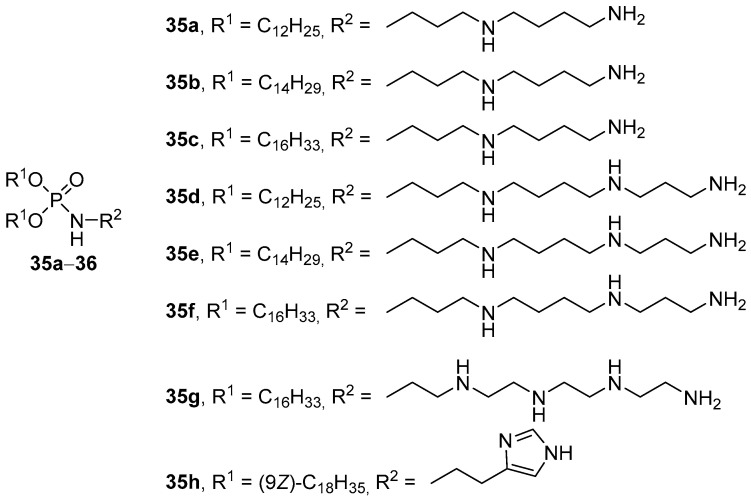
Phosphamide derivatives of polyamines.

**Figure 11 pharmaceutics-13-00920-f011:**
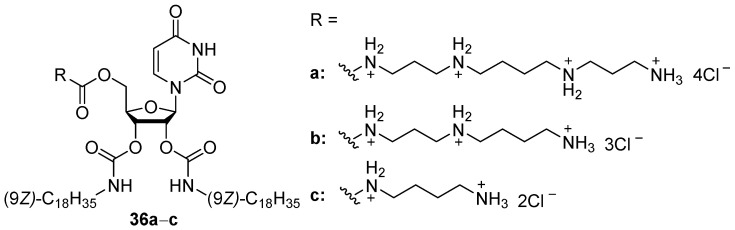
Cationic nucleoside amphiphiles.

**Figure 12 pharmaceutics-13-00920-f012:**
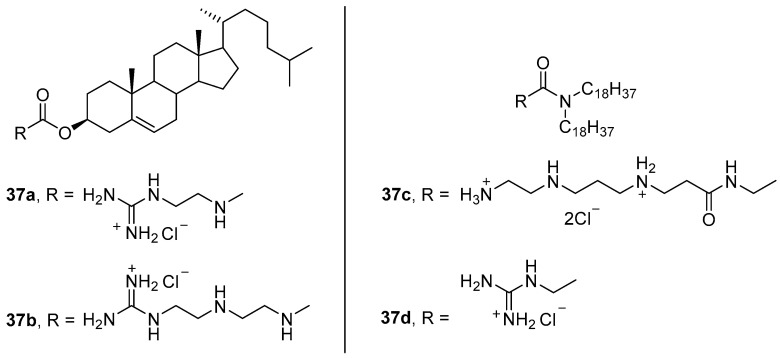
Amphiphiles based on polyamines and cholesterol.

**Figure 13 pharmaceutics-13-00920-f013:**
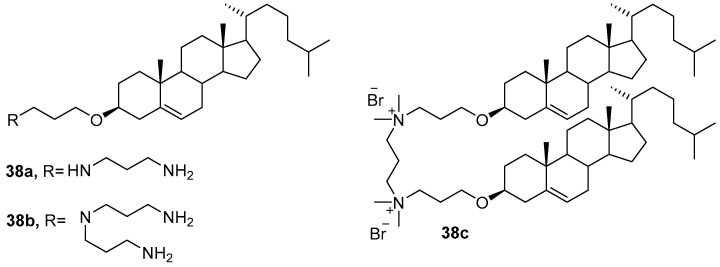
Ether-linked cationic amphiphiles.

**Figure 14 pharmaceutics-13-00920-f014:**
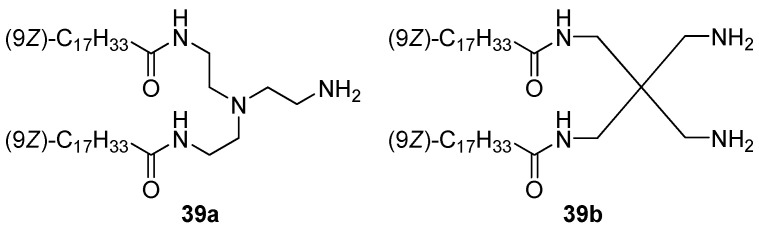
Branched cationic amphiphiles.

**Figure 15 pharmaceutics-13-00920-f015:**
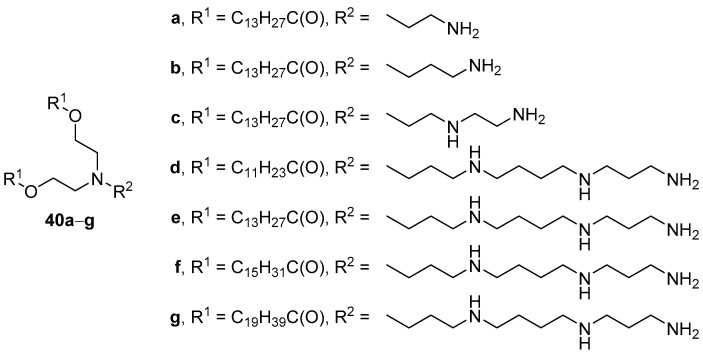
CAs based on various polyamines.

**Figure 16 pharmaceutics-13-00920-f016:**
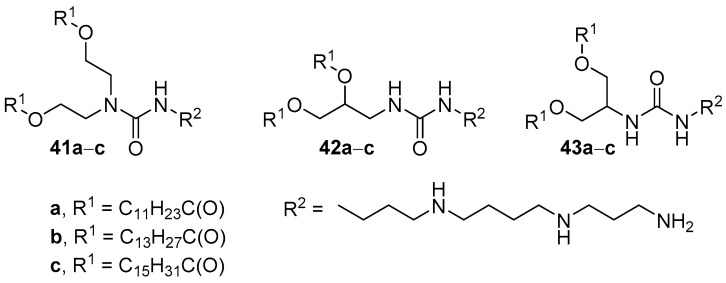
Spermine-based CAs with different cores.

**Figure 17 pharmaceutics-13-00920-f017:**
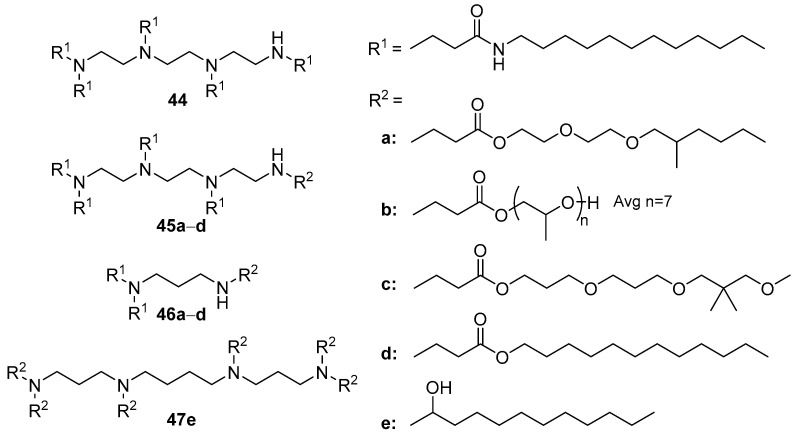
A combinatorial library of CAs.

**Figure 18 pharmaceutics-13-00920-f018:**
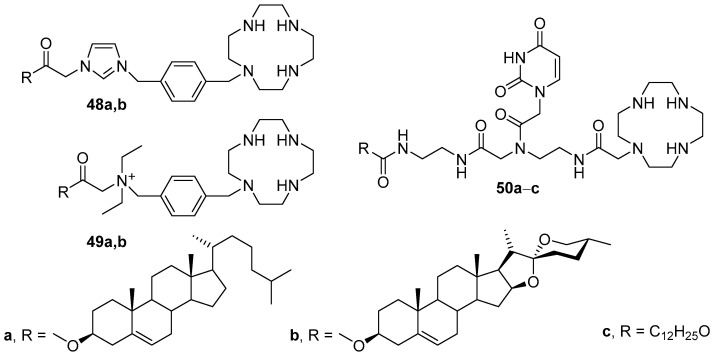
Cyclen-based amphiphiles.

**Figure 19 pharmaceutics-13-00920-f019:**
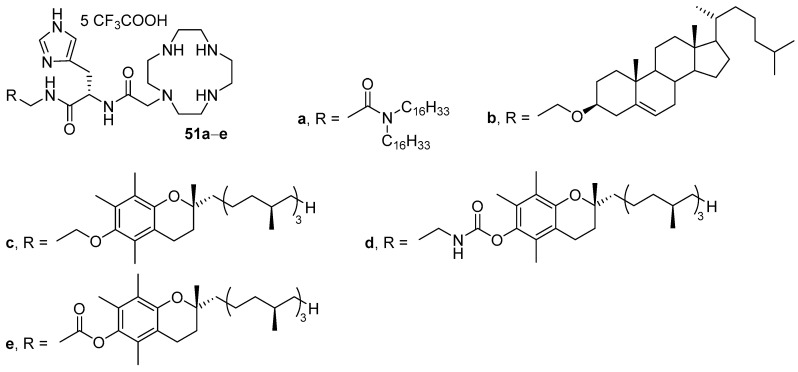
Cyclen-based amphiphiles with l-histidine backbone.

**Figure 20 pharmaceutics-13-00920-f020:**
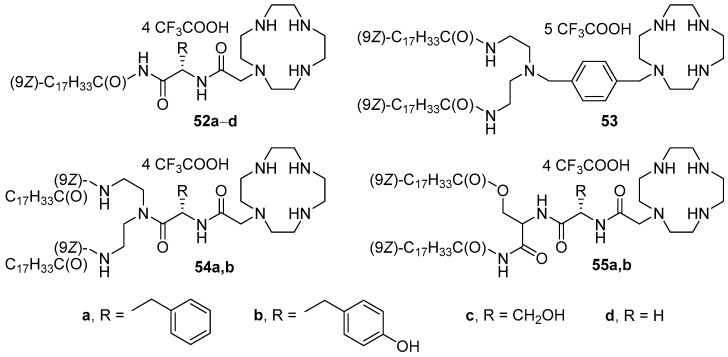
Cyclen-based amphiphiles with different amino acid backbones.

**Figure 21 pharmaceutics-13-00920-f021:**
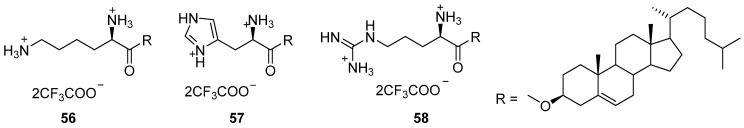
CAs based on amino acids and cholesterol containing no spacers.

**Figure 22 pharmaceutics-13-00920-f022:**
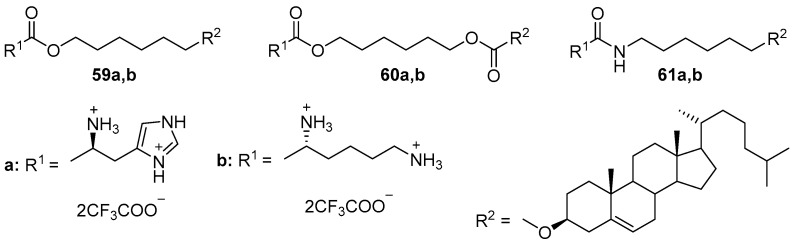
CAs based on amino acids and cholesterol with various linkers.

**Figure 23 pharmaceutics-13-00920-f023:**
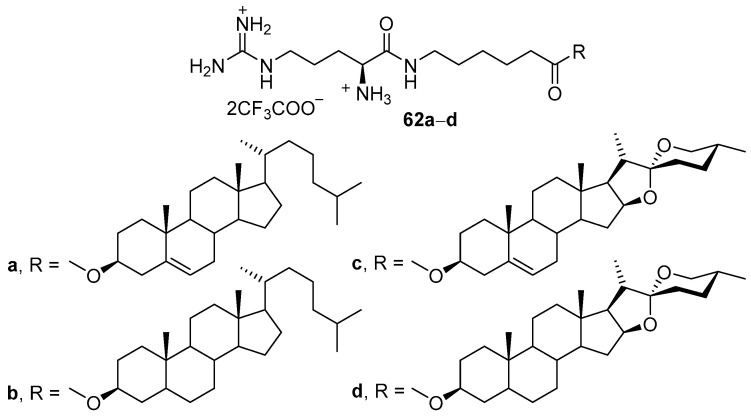
CAs based on l-arginine and various sterols.

**Figure 24 pharmaceutics-13-00920-f024:**
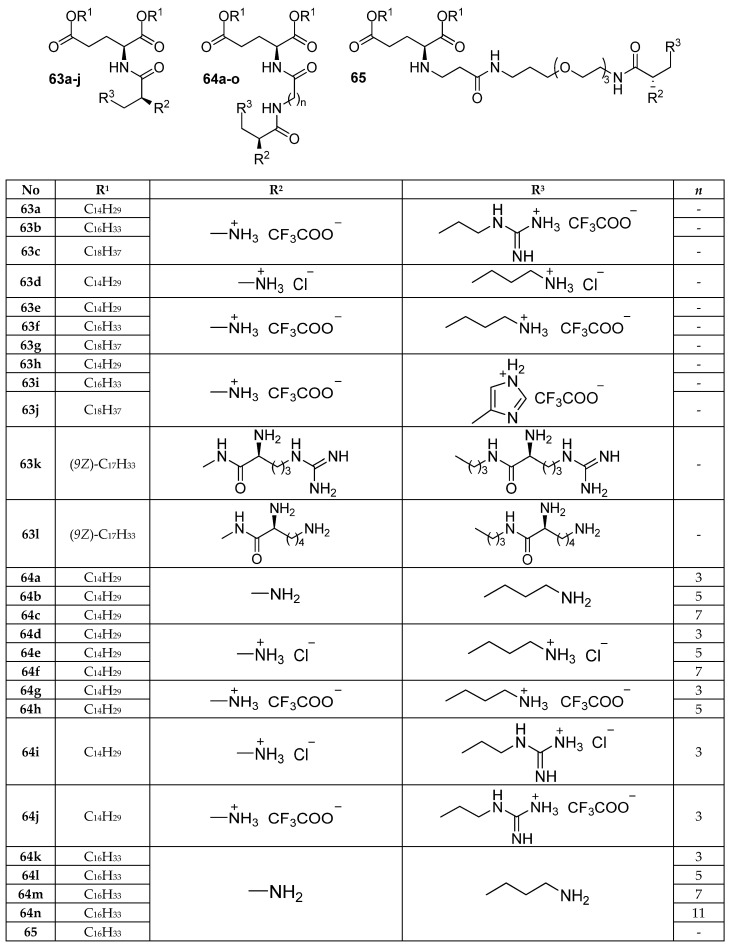
CAs based on l-arginine, l-lysine, and l-histidine.

**Figure 25 pharmaceutics-13-00920-f025:**
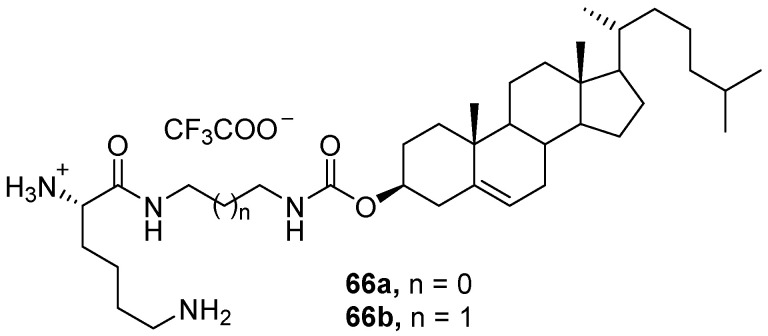
CAs based on l-lysine and diamines.

**Figure 26 pharmaceutics-13-00920-f026:**
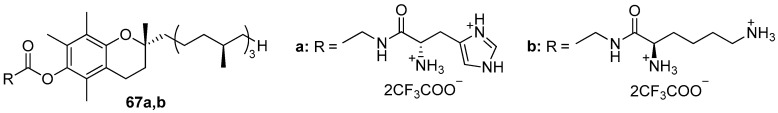
CAs based on dipeptides.

**Figure 27 pharmaceutics-13-00920-f027:**
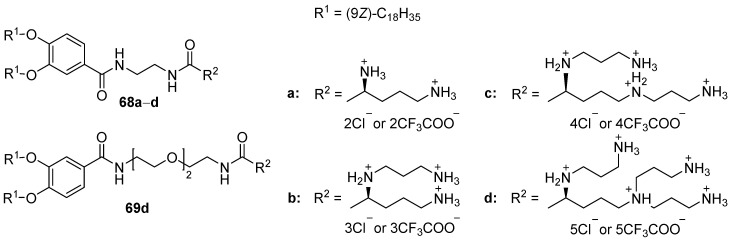
CAs based on L-ornithine.

**Figure 28 pharmaceutics-13-00920-f028:**
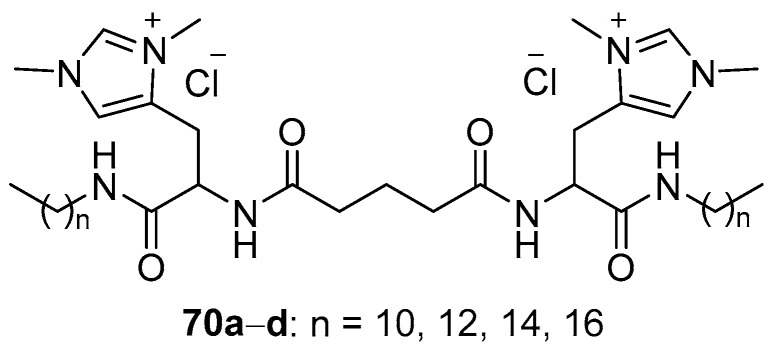
Gemini-amphiphiles based on L-histidine.

**Figure 29 pharmaceutics-13-00920-f029:**
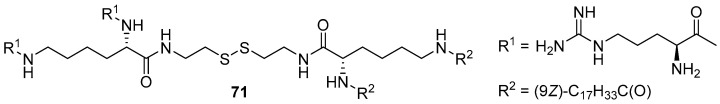
Disulfide amphiphiles based on l-lysine and l-arginine.

**Figure 30 pharmaceutics-13-00920-f030:**
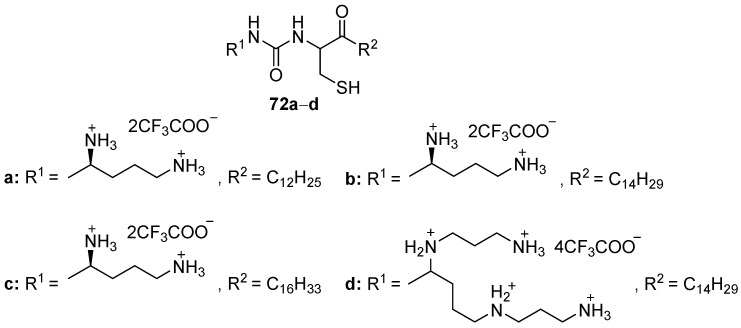
Disulfide amphiphiles based on l-ornithine or spermine.

**Figure 31 pharmaceutics-13-00920-f031:**
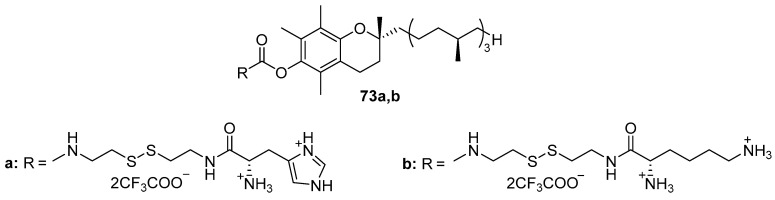
Disulfide amphiphiles based on tocopherol.

**Figure 32 pharmaceutics-13-00920-f032:**
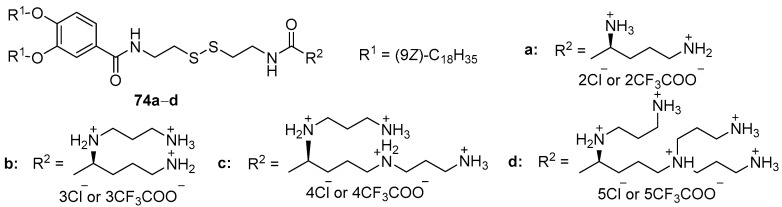
Disulfide amphiphiles based on different polyamines.

**Figure 33 pharmaceutics-13-00920-f033:**
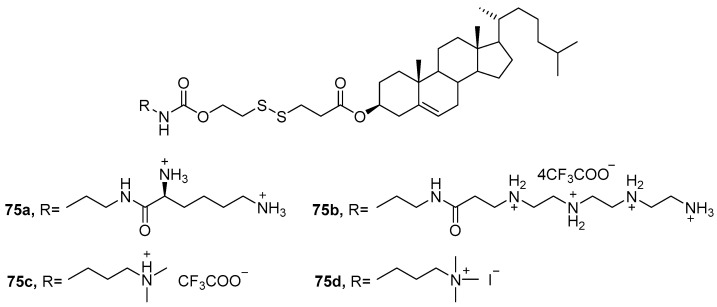
Disulfide amphiphiles based on polyamines, amino acids, and cholesterol.

**Figure 34 pharmaceutics-13-00920-f034:**
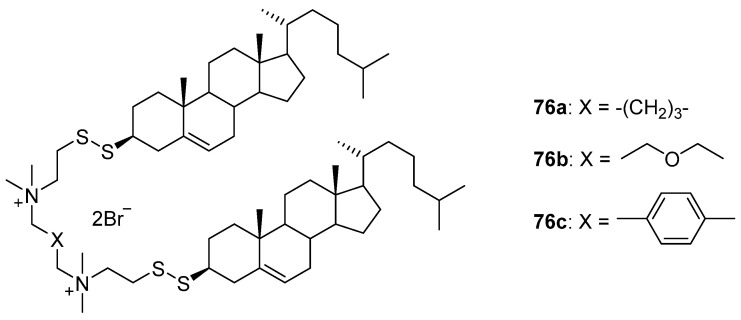
Disulfide gemini-amphiphiles with different spacers.

**Figure 35 pharmaceutics-13-00920-f035:**
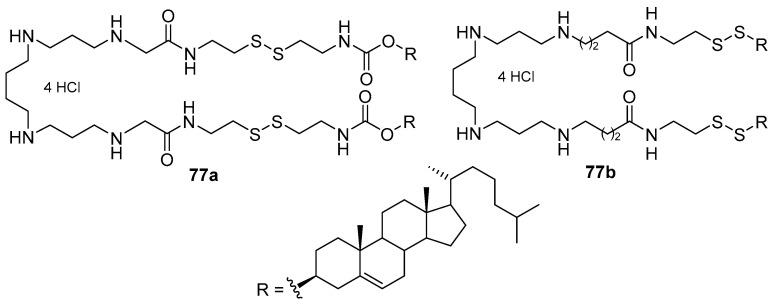
Disulfide polycationic amphiphiles based on spermine and cholesterol.

**Figure 36 pharmaceutics-13-00920-f036:**
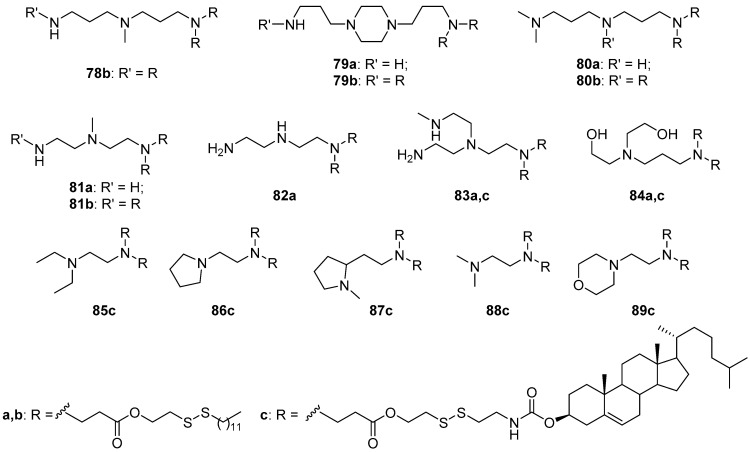
A library of disulfide CAs with various cationic domains.

## Data Availability

Not applicable.

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
