# Peer review of "Lipophilic Polyamines as Promising Components of Liposomal Gene Delivery Systems"

_pharmaceutics, 2021, doi:10.3390/pharmaceutics13060920_

Round 1

Reviewer 1 Report

This paper presents a comprehensive review on the structures of polycationic amphiphiles that were synthesized for the purpose of nucleic acid delivery.

I recommend the authors to revise the manuscript for the following aspects before it’s publication.

  1. Line 20-22: consider to delete ‘including COVID19’ or modify the sentence to make it proper. As it is described later in the next paragraph, use of liposome in COVID19 vaccine is not an example of delivering a damaged gene or nucleic acid to block the expression of a gene (siRNA).
  2. The introduction of general compositions of liposomes in the introduction is great (line 42-47). However, later in the manuscript, whether each cationic amphiphile was used with or without any additional helper lipids to form liposomes is not that clear. In many examples, DOPE was clearly mentioned as an additional component of liposomal formulations (for example lines 84-86, 139, 177, 185, 193-197, 208-21, 280, 383, 392, 457). But most of the other examples, there are no mention about the rest of the formulations. Is that because the cationic amphiphile is the sole component in those examples? If that’s the case, the authors should make it clear in the introduction as a guidance for the readers. If that’s not the case, I highly suggest that the authors to develop a good way of summarizing the other components of each formulation throughout the manuscript.
  3. Cytotoxicity of the liposomes could origin from multiple reasons. However, the discussion regarding toxicity is done too liberally without properly suggested rationale. For example, some amphiphiles can be toxic because they disrupt the membranes too much, while the others could be toxic because of their non-degradability within the endo-lysosomes. If not discussed in each example, the authors can elaborate the relationship between the structure of cationic amphiphiles and cytotoxicity more in Sec 2.5.
  4. In line 413-414, the authors are introducing protein reductidases and glutathione as potential mechanism for reduction of disulfide bonds. Is that truly the case within the cells, i.e. cytoplasm and endosomes? How about the change in pH?
  5. In line 57-60, the polyamine transporter is introduced as a main receptor and mechanism behind transfection using amine-containing formulations. Is that true? Isn’t it generally accepted that the charge-charge interaction between the negatively-charged plasma membrane and the positively-charged liposome formulations increases as the charge density (amine density) goes up? Or is it related to the pKa values of primary, secondary, or tertiary amines present in the molecules? These aspects should be included in the manuscript more carefully.

Reviewer 2 Report

The manuscript entitled “Lipophilic Polyamines as Promising Components of Liposomal Gene Delivery Systems” is about summarizing both strategies and polyamine derivatives that have been used to deliver plasmids, oligonucleotides not only in cell culture but also in vivo. This revision is excellent, well-organized, and timely.

In conclusion, my opinion is totally positive so that I strongly support the publication of this significant manuscript in Pharmaceutics.

Minor points

  • Examples of polyamine-conjugated cationic nucleoside lipids have been reported in the literature for siRNA delivery. This might be also included in this manuscript.
  • The authors should include a conclusion section including future perspectives

Round 2

Reviewer 1 Report

The authors have diligently addressed the questions and concerns generated by me. Thank you.